# Novel Sensing Technique for Stem Cells Differentiation Using Dielectric Spectroscopy of Their Proteins

**DOI:** 10.3390/s23052397

**Published:** 2023-02-21

**Authors:** Young Seek Cho, So-Jung Gwak

**Affiliations:** 1Department of Electronic Engineering, Wonkwang University, Iksan 54538, Jeollabuk-do, Republic of Korea; 2Department of Chemical Engineering, Wonkwang University, Iksan 54538, Jeollabuk-do, Republic of Korea

**Keywords:** complex permittivity spectra, dielectric spectroscopy, dielectrophoresis, immunohistochemistry, open-ended coaxial probe, protein, stem cell

## Abstract

Dielectric spectroscopy (DS) is the primary technique to observe the dielectric properties of biomaterials. DS extracts complex permittivity spectra from measured frequency responses such as the scattering parameters or impedances of materials over the frequency band of interest. In this study, an open-ended coaxial probe and vector network analyzer were used to characterize the complex permittivity spectra of protein suspensions of human mesenchymal stem cells (hMSCs) and human osteogenic sarcoma (Saos-2) cells in distilled water at frequencies ranging from 10 MHz to 43.5 GHz. The complex permittivity spectra of the protein suspensions of hMSCs and Saos-2 cells revealed two major dielectric dispersions, β and γ, offering three distinctive features for detecting the differentiation of stem cells: the distinctive values in the real and imaginary parts of the complex permittivity spectra as well as the relaxation frequency in the β-dispersion. The protein suspensions were analyzed using a single-shell model, and a dielectrophoresis (DEP) study was performed to determine the relationship between DS and DEP. In immunohistochemistry, antigen–antibody reactions and staining are required to identify the cell type; in contrast, DS eliminates the use of biological processes, while also providing numerical values of the dielectric permittivity of the material-under-test to detect differences. This study suggests that the application of DS can be expanded to detect stem cell differentiation.

## 1. Introduction

Dielectric spectroscopy (DS) characterizes the rotation and relaxation of dipole molecules in solid, liquid, or gas phases in the presence of an external electric field applied to materials [1]. By studying the motion of dipoles in a material-under-test (MUT), DS presents the complex permittivity spectra of the MUT over a broadband frequency range.

Since the pioneering work of H. P. Schwan, who presented the electrical properties of tissues and cell suspensions from 10 Hz to 35 GHz [2], DS has been the primary technique used for investigating the dielectric properties of biological materials. There are three major dispersion types: α, β, and γ. The α-dispersion is difficult to measure because of interference from electrode polarization but is usually found below 10 kHz. The β-dispersion was mainly due to interfacial polarization and was attributed to the existence of the cell plasma membrane. The γ-dispersion is attributed to the relaxation of the water molecules. β- and γ-dispersions occur in the radio frequency (RF) band (MHz range) and microwave frequency band (GHz range), respectively.

The dielectric properties of many human tissues, including the blood, bone, brain, fat, and heart have been previously characterized by Gabriel et al. [3,4,5]. The tissues investigated had a frequency range of 10 Hz–20 GHz. Open-ended coaxial probes with an impedance analyzer (10 Hz–10 MHz) and two network analyzers (300 kHz–3 GHz and 130 MHz–20 GHz) were used to measure the human tissues.

Subsequently, DS was used to investigate the dielectric properties at the celluar level [6,7,8,9]. Asami et al. simulated and measured mouse erythrocytes, lymphocytes, and plant protoplasts at frequencies between 10 kHz and 250 MHz. One dispersion was observed at approximately 5 MHz in mouse erythrocytes, and two dispersions were observed in mouse lymphocytes: a large dispersion at approximately 1 MHz and a small dispersion at approximately 6 MHz.

The DS of human erythrocyte suspensions was performed by Bao et al. over low frequency ranges [10,11,12]. An impedance measurement technique developed in-house was used for the low-frequency range (1 Hz–10 MHz), and an open-ended coaxial probe with an automated network analyzer was used for the RF/microwave frequency. β- and γ-dispersions were observed in human erythrocyte suspensions in the frequency range of 45 MHz to 26.5 GHz.

Following the investigations of tissues and cells, DS has been used to analyze the dielectric properties of DNA. The dielectric properties of calf thymus DNA dissolved in water have been characterized in the frequency range of 10 MHz to 10 GHz [13]. The complex permittivity spectra of salmon sperm DNA solutions were measured from 25 MHz to 110 GHz by Ermilova et al. [14]. An open-ended coaxial probe with a vector network analyzer (VNA) was used for the analysis. β- and γ-dispersions were observed at approximately 100 MHz and 19 GHz, respectively, and a small dispersion, called the δ-dispersion, was observed at approximately 300 MHz with high DNA concentrations. Ermilova et al. measured and characterized the dielectric properties of biomolecules in ionic solutions up to 110 GHz [15].

DS has drawn significant attention for its clinical use in dielectric sensing and the detection of biomaterials in solid and liquid phases. The noninvasive, label-free, and real-time sensing characteristics of this technique make it ideal for clinical use. The dielectric properties of normal and diseased breast tissues were determined using an open-ended coaxial probe over a frequency range of 1–20 GHz [16]. Lymphatic filariasis, a parasitic infection spread by mosquitoes, can be detected using DS measurements of the patient’s blood [17]. A millimeter-wave DS imaging system for skin cancer was developed using an open-ended coaxial probe [18]. The authors’ previous work also demonstrated the feasibility of detecting a virus using DS [19].

Numerous methods for specific protein expression in cells, such as immunostaining, Western blotting, flow cytometry, microarrays, immunocytochemistry, quantitative polymerase chain reaction (qPCR), and reverse transcription polymerase chain reaction (RT-PCR), have been used to characterize cell differentiation [20,21]. Specifically, immunocytochemistry is the most commonly used fluorescence method, in which secondary antibodies with fluorescent tags detect the primary antibody bound to the target protein for the characterization of cells [22,23,24,25,26]. However, the resolution of these techniques is limited, making them unsuitable for the highly sensitive characterization of stem cells.

Several electrical detection techniques have been reported for the detection of stem cell differentiation. Using electrode-based chip and impedance spectroscopy, [27] investigated the influence of chlorpyrifos (an insecticide) on the differentiation of hMSCs to adipocytes. Moreover, the neural differentiation of hMSCs can be monitored by measuring the cellular electrical resistance using electrical cell-substrate impedance sensing [28]. Although DS was used to characterize human fetal osteoblastic and osteosarcoma cell suspensions in the frequency range of 500 MHz to 10 GHz using an open-ended coaxial probe [29] or human-induced pluripotent stem cells using a capacitance sensor below 10 MHz [30], it has rarely been used to characterize stem cells in the broadband frequency range.

In this study, we propose the use of DS, using an open-ended coaxial cable with a VNA, as a sensing technique to distinguish the differentiation of stem cells, hMSCs, and Saos-2 cells in distilled water (DW). The complex permittivity spectra of the protein suspension of hMSCs and Saos-2 cells were presented in the frequency range of 10 MHz to 43.5 GHz. The complex permittivity spectra clearly showed β- and γ-dispersion and three other distinctive features: the real and imaginary parts of the complex permittivity spectra as well as the relaxation frequency in β-dispersion. These features distinguished hMSCs from Saos-2 cells. Immunohistochemistry, the most popular technique, was used to characterize hMSCs and Saos-2 cells.

Besides DS of protein suspensions, we also investigate the connection between DS and dielectrophoresis (DEP) of proteins. Recent studies for DEP of solvated proteins indicated that the standard DEP theory, based on the macroscopic Clausius–Mossotti factor, needed to be replaced by a new theory that models cross-correlated dipole interactions between protein dipole and water dipoles in its hydration shell [31,32]. Colburn et al. demonstrated that protein could be trapped on a nanopore with by DEP force [33]. In this study, using the β-dispersion parameters of the protein suspensions of hMSCs and Saos-2 cells from DS, it will be shown that DEP can discriminate the proteins from hMSCs and Saos-2 cells.

This article is organized as follows: Section 2 presents the preparation of stem cells and the DS experimental methodology. Section 3 describes the experimental results from biological and electrical measurements followed by Section 4 for discussion for this work. Finally, the conclusions are presented in Section 5.

## 2. Preparation of Biological Samples and Experimental Setup

### 2.1. Preparation of Biological Samples

#### 2.1.1. Cell Culture

Human mesenchymal stem cells (hMSCs, ATCC, Manassas, VA, USA) and human osteogenic sarcoma cells (Saos-2 cells; Korea Cell Line Bank, Seoul, Republic of Korea) were cultured in Dulbecco’s modified Eagle’s medium/Ham’s F-12 50/50 (DMEM/F12; Gibco, New York, NY, USA) supplemented with 10% fetal bovine serum (FBS; Gibco), 100 U/mL penicillin, and 100 μg/mL streptomycin (Gibco) in a humidified incubator at 37 ${}^{\circ }C$ with 5% CO_2_. The medium was changed on alternate days.

#### 2.1.2. Sample Preparation for Dielectric Spectroscopy

The cells were detached from the cell culture dish by trypsinization, and the cell suspension was centrifuged at 1500 rpm for 3 min. The supernatant was removed, and the cells were lysed using a homogenizer in DW. To remove cell debris, the sample was transferred to a microtube and centrifuged at 10,000 rpm for 10 min. After centrifugation, the protein suspensions of hMSCs or Saos-2 cells were collected in new microtubes. The total protein in the protein suspensions was measured using the SMART BCA Protein Assay Reagent Kit (iNtRon Biotechnology, Inc., Seongnam-Si, Republic of Korea)

#### 2.1.3. Cell Characterization by Immunohistochemistry

For immunofluorescence, cells were cultured on cell culture slides (SPL Life Sciences Co., Ltd., Pocheon-si, Gyeonggi-do, Republic of Korea) fixed with 4% (*v*/*v*) paraformaldehyde at room temperature. After fixation, the cells were stained with primary antibodies against CD 90 (Santa Cruz Biotechnology, Dallas, TX, USA), CD 73 (Santa Cruz Biotechnology) or osteocalcin (Santa Cruz Biotechnology). CD 90 and CD 73 are expressed in hMSCs and are used as phenotypic markers of mesenchymal stem cells. Osteocalcin is expressed in Saos-2 cells and used as an osteoblast and bone marker.

The cells were stained using fluorescein 5-isothiocyanate(FITC)-conjugated anti-mouse IgG secondary antibodies (Jackson ImmunoResearch Laboratories Inc., West Baltimore Pike, PA, USA). Vectashield® mounting medium with 4’,6-diamidino-2-phenylindole (DAPI; Vector Laboratories, Newark, CA, USA) was used to stain the cell nuclei. The stained samples were analyzed by fluorescence microscopy.

### 2.2. Experimental Setup for Dielectric Spectroscopy

A previous methodology [19] was updated for use in this study (a schematic is shown in Figure 1). A drawing of the open-ended coaxial probe, 200 mm long and assembled with a 2.4 mm male coaxial connector, is shown in Figure 1a. One end of the probe was connected to the VNA using a coaxial cable. The other end was submerged in an aqueous biological material within a 2.0 mL microtube, as shown in Figure 1b.

The measurement setup is illustrated in Figure 2. A coaxial cable, assembled with 2.4 mm female coaxial adapters on both ends, was connected to the open-ended coaxial probe and VNA. The open-ended coaxial probe was the “Slim form probe” (Keysight Technologies, Santa Rosa, CA, USA). The VNA (N5224B PNA Microwave Network Analyzer, Keysight Technologies) covered the frequency band between 10 MHz and 43.5 GHz.

After connecting the probe and VNA, the reference plane for measurement was shifted from the VNA test port to the end of the probe via VNA calibration. Shifting the reference plane allowed for the removal of the interface between the probe and the coaxial cable, as well as the coaxial cable and 200 mm long probe, from the measured one-port S-parameters ($S_{11}$). Calibration of the VNA was performed in three steps. Firstly, the probe was freely hung in midair, called “open circuit”. Secondly, the probe was connected to a “Slim form short” provided by the manufacturer, called “short circuit”. Thirdly, the probe was submerged in a 25 mL glass vial of DW at room temperature, called “load”. During the calibration process and measurements of the MUT, the entire VNA frequency band (10 MHz–43.5 GHz) was swept on a logarithmic scale over 401 frequency data points to measure $S_{11}$. The $S_{11}$ was averaged over 16 measurements.

From the $S_{11}$ measured, a software (Keysight N1500A Materials Measurement Suits—Coaxial probe method—pre-installed in the VNA) was used to extract the complex permittivity spectra. When the calibration process was completed, the spectra of a second 25 mL glass vial of DW were extracted to validate the calibration process. The complex permittivity spectra of DW are presented in Section 3.

## 3. Experimental Results

The experimental results are divided into two categories: the first is biological characterization and the second is electrical characterization. In Section 4, the interpretation and experimental conclusions drawn from both characterizations are presented.

### 3.1. Biological Characterizations

Immunohistochemistry (IHC) detects antigens or proteins in cells or tissue sections by binding specific antibodies to the antigens of interest. IHC is widely used in research and clinical laboratories to visualize the distribution and localization of specific cellular components, such as proteins and other macromolecules, within cells and tissues. Different cells have different proteins that can be identified by antigen–antibody interactions using IHC.

In this study, hMSCs and Saos-2 cells were characterized by immunofluorescence using CD 90, CD 73, and osteocalcin to determine different cell types (Figure 3). The hMSCs expressed mesenchymal stem cell markers CD 90 and CD 73. Saos-2 cells, however, did not express either marker. Merging of marker and DNA images showed that most hMSCs were positive for CD 90 and CD 73, confirming that hMSCs have the expected stem cell characteristics. Alternatively, Saos-2 cells expressed the osteoblast-specific marker osteocalcin (Figure 3c). Merging of marker and DNA images showed that all Saos-2 cells were stained with osteocalcin, indicating that osteoblast properties were maintained.

The hMSCs are multipotent cells that are capable of differentiating into various specialized tissue cells, including osteoblasts, chondrocytes, and adipocytes. Quality control of stem cell differentiation is critical for clinical therapy, which requires the accurate evaluation of cell surface markers and the expression of molecules. Stem cell markers, such as CD 90, CD 73, and CD 105, are commonly used to isolate and identify stem cells. Currently, the industry requires the characterization of hMSCs using CD 73, CD 90, CD 105, and CD 44. Positive hMSCs have been used for tissue regeneration [34,35]. The cultured hMSCs express CD 105, CD 73, and CD 90 but do not express CD 31, CD 14, or mature markers of tissue-specific cells [36]. Saos-2 cells have been widely used in studies of bone cell differentiation, proliferation, and metabolism and are known to be capable of bone production. They exhibit the most mature osteoblastic phenotype and are positive for alkaline phosphatase, osteocalcin, and collagen I and III [37].

### 3.2. Electrical Characterizations

#### 3.2.1. Measurements Results

The extracted real ($\varepsilon_{r}^{\prime }$) and imaginary ($\varepsilon_{r}^{\prime \prime }$) parts of the complex permittivity spectra of the DW are shown in Figure 4a and Figure 4b, respectively. The $\varepsilon_{r}^{\prime }$ of DW remained around 77–78 to 3 GHz and decreased above 3 GHz because of the γ-dispersion, as expected. The measured value was close to the known static permittivity of water of 78.35±0.05 [38]. The maximum value of $\varepsilon_{r}^{\prime \prime }$ was also observed at 19.6 GHz, which is the relaxation frequency ($f_{c,\gamma }$) of DW in the γ-dispersion. In general, the $f_{c}$ of a dielectric can be converted to the relaxation time (τ) using the relationship given in Equation (Equation 1).
(1)$$\tau =\frac{1}{\omega_{c}}=\frac{1}{2\pi f_{c}}$$

The $\tau_{\gamma }$ value of DW in the γ-dispersion was 8.12 ps. This value is close to the value of 8.27±0.02 ps previously reported [38]. The DW characteristics of $\varepsilon_{r}^{\prime }$ and $\varepsilon_{r}^{\prime \prime }$ over the frequency band of 10 MHz to 43.5 GHz confirm that the experimental setup (Figure 2) is valid for measuring and characterizing the complex permittivity spectra of aqueous biological materials.

The $\varepsilon_{r}^{\prime }$ spectra of the protein suspensions of hMSCs and Saos-2 cells were compared with each other and with that of DW (Figure 4a). The $\varepsilon_{r}^{\prime }$ spectra above 300 MHz were quite similar for the three aqueous solutions and were difficult to differentiate. However, differences occurred below 300 MHz; the $\varepsilon_{r}^{\prime }$ values of both protein types were greater than those of DW as the frequency decreased. At 30 MHz, the $\varepsilon_{r}^{\prime }$ of the protein suspension of hMSCs was 94.4, which was 7.4 and 15.7 higher than that of the protein suspension of Saos-2 cells and DW, respectively. The increase in the $\varepsilon_{r}^{\prime }$ spectra of the protein suspensions of hMSCs and Saos-2 cells was caused by interfacial polarization, which is known as the Maxwell–Wagner effect. Interfacial polarization occurs when a dielectric particle is placed in an ionic solution or electrolyte and an electric field is applied to the solution [39]. This phenomenon is also known as β-dispersion and is typically exhibited in biomaterial solutions in the RF range, usually less than 1 GHz [2].

The $\varepsilon_{r}^{\prime \prime }$ spectra of the protein suspensions of hMSCs and Saos-2 cells were compared with each other and to DW (Figure 4b). The $\varepsilon_{r}^{\prime \prime }$ spectra above 1 GHz were similar for the three samples and could not be differentiated. However, the samples could be differentiated at frequencies below 1 GHz; the $\varepsilon_{r}^{\prime \prime }$ values of the suspensions became greater than DW as the frequency decreased. At 30 MHz, the $\varepsilon_{r}^{\prime \prime }$ of the protein suspension of hMSCs was 28.7, which was 10.8 and 24.4 higher than that of the protein suspension of Saos-2 cells and DW, respectively. The increase in the $\varepsilon_{r}^{\prime \prime }$ spectra of the protein suspensions of hMSCs and Saos-2 cells was mainly due to the conductivity of the solution, with the protein suspension of hMSCs having a greater conductivity than the protein suspension of Saos-2 cells.

The $\varepsilon_{r}^{\prime }$ and $\varepsilon_{r}^{\prime \prime }$ values of the three materials were compared at specific frequencies (Table 1). Because $\varepsilon_{r}^{\prime }$ and $\varepsilon_{r}^{\prime \prime }$ differ between the protein suspensions of hMSCs and Saos-2 cells below 300 MHz and 1 GHz, respectively, it is possible to differentiate stem cells based on their complex permittivity spectra.

In addition to the β-dispersion, γ-dispersion was observed among the three materials. The γ-dispersion originates from the orientation polarization of water molecules in the bulk phase and was measured in the microwave frequency range. Orientation polarization occurs in a dielectric consisting of permanent dipoles, such as water, when an electric field is applied [39]. The $\varepsilon_{r}^{\prime \prime }$ spectra of the three materials (Figure 4b) focused on the microwave frequency range (Figure 5). The relaxation frequencies of the protein suspensions of hMSCs and Saos-2 cells were different from those of DW (Table 2). In the presence of stem cell proteins, the $f_{c,\gamma }$ of DW in the γ-dispersion was affected slightly, moving 2.3 and 1.9 GHz to a lower frequency band in the protein suspensions of hMSCs and Saos-2 cells, respectively. Although the $f_{c,\gamma }$ value of the protein solution of stem cells in the γ-dispersion is distinguishable from that of DW, the $f_{c,\gamma }$ values of the protein suspensions of hMSCs and Saos-2 cells is similar. As a result, the $f_{c,\gamma }$ of stem cells in the γ-dispersion cannot be used to differentiate stem cells, whereas $f_{c,\beta }$ in the β-dispersion can.

#### 3.2.2. Single Shell Model for Protein Suspensions

DS provides complex permittivity spectra over the frequency band of interest. The spectra can be expressed as a function of frequency as follows: (2)$$\begin{array}{c} \varepsilon_{r}\left(\omega \right)=\varepsilon_{r}^{\prime }\left(\omega \right)-j\varepsilon_{r}^{\prime \prime }\left(\omega \right) \end{array}$$
where ω=2πf and *f* is the frequency. The $\varepsilon_{r}^{\prime }\left(\omega \right)$ of a material represents the relative electrical energy storage capability of the material compared to free space. The $\varepsilon_{r}^{\prime \prime }\left(\omega \right)$ of the material represents the attenuation of electromagnetic waves passing through the material [40].

Because of the limitation of VNA bandwidth in the low frequency band of 10 MHz in this study, it is difficult to obtain an overall picture of the dielectric dispersion characteristics, especially in the kHz band. However, it may be possible to draw an overall picture of the dielectric dispersion characteristics using a mathematical model for a cell or protein suspension and a curve-fitting technique.

The dielectric properties of a cell suspension can be analyzed using the “multi-stratified shell” model developed by Irimajiri et al. [41,42,43]. The “multi-stratified shell” model can be expressed as follows: (3)$$\begin{array}{c} \varepsilon_{r}^{*}\left(\omega \right)=\varepsilon_{r,\infty }+\sum_{k=1}^{n+1}\frac{\Delta \varepsilon_{k}}{1+j\omega \tau_{k}}+\frac{\sigma_{dc}}{j\omega \varepsilon_{0}} \end{array}$$
where $\varepsilon_{r,\infty }$ is the dielectric constant at f=∞, $\Delta \varepsilon_{k}$ is the dielectric decrements in the *k*th “unit” dispersion, $\tau_{k}$ is the relaxation time in the *k*th “unit” dispersion, respectively, and $\sigma_{dc}$ and $\varepsilon_{0}$ are the DC conductivity of the suspension and electrical permittivity in free space, respectively. According to the “multi-stratified shell” model, the number of the dielectric “unit” dispersion corresponds to the number of interfaces between the dielectrics in the suspension.

When the protein suspension is made with water, there is a layer or two of water molecules (hydration shell) strongly associated with the protein surface [44], known as “bound water” [45]. As explained in Section 3.2.1, two distinct dielectric dispersions, called β- and γ-dispersions, were observed in the protein suspensions of hMSCs and Saos-2 cells. The β-dispersion occurs because the “bound water” acts as a “single-shell” for the protein. Therefore, with n=1 (called “single-shell” model in [42]) in Equation (Equation 3), the dielectric properties of the protein solution of hMSCs and Saos-2 cells can be modeled with Equation (Equation 4) as follows: (4)$$\begin{array}{c} \varepsilon_{r}^{*}\left(\omega \right)=\varepsilon_{r,\infty }+\frac{\Delta \varepsilon_{\beta }}{1+j\omega \tau_{\beta }}+\frac{\Delta \varepsilon_{\gamma }}{1+j\omega \tau_{\gamma }}+\frac{\sigma_{dc}}{j\omega \varepsilon_{0}}. \end{array}$$

The first dispersion (k=1 in Equation (Equation 3)) and the second dispersion (k=2 in Equation (Equation 3)) correspond to β- and γ-dispersion in Equation (Equation 4), respectively.

By using a complex nonlinear least squares fit [12,14], the unknown parameters—$\varepsilon_{r,\infty }$, $\Delta \varepsilon_{\beta }$, $\Delta \varepsilon_{\gamma }$, $\tau_{\beta }$, and $\sigma_{dc}$ in Equation (Equation 4)—can be determined. In this study, a commercially available software tool (OriginPro 2021b, OriginLab Corporation, Northampton, MA, USA) using the Levenberg–Marquardt algorithm was used to determine the unknown parameters.

The “single-shell” model parameters between the protein suspensions of hMSCs and Saos-2 cells were compared (Table 3). The curve-fitted complex permittivity spectra ($\varepsilon_{r}^{*}\left(\omega \right)$) in the extended frequency band (100 kHz to 1 THz) are compared with the measured complex permittivity spectra ($\varepsilon_{r}\left(\omega \right)$) (Figure 6). In the extended frequency band, it may be possible to present an overall picture of the dielectric dispersion of the protein suspensions of hMSCs and Saos-2 cells.

The $\varepsilon_{r}^{\prime }$ spectra of the curve-fitted $\varepsilon_{r}^{*}\left(\omega \right)$ were in good agreement with those of the measured $\varepsilon_{r}^{\prime }\left(\omega \right)$ (Figure 6a). The “single-shell” model (Equation Equation 4) predicts the difference between the protein solution of hMSCs and Saos-2 cells, which is 57.134 at 100 kHz. However, when f=∞, the difference in $\varepsilon_{r,\infty }$ is negligible (Table 3).

As mentioned earlier, $\varepsilon_{r}^{\prime \prime }\left(\omega \right)$ represents the attenuation of electromagnetic waves passing through the material. The larger the $\varepsilon_{r}^{\prime \prime }\left(\omega \right)$ for a dielectric, the more is the loss for the dielectric. In fact, the $\varepsilon_{r}^{\prime \prime }\left(\omega \right)$ spectra can be expressed [12,14,15] as: (5)$$\varepsilon_{r}^{\prime \prime }\left(\omega \right)=\varepsilon_{rd}^{\prime \prime }\left(\omega \right)+\varepsilon_{r\sigma }^{\prime \prime }\left(\omega \right)=\varepsilon_{rd}^{\prime \prime }\left(\omega \right)+\frac{\sigma_{dc}}{\omega \varepsilon_{0}}$$
where $\varepsilon_{rd}^{\prime \prime }\left(\omega \right)$ and $\varepsilon_{r\sigma }^{\prime \prime }\left(\omega \right)$ represent the dielectric and conductor losses, respectively. The $\varepsilon_{r\sigma }^{\prime \prime }\left(\omega \right)$ can be expressed as the absolute value of the last term in Equation (Equation 4), as shown in Equation (Equation 5). Because the $\varepsilon_{r}^{\prime \prime }\left(\omega \right)$ in the β-dispersion region is dominated by the last term, $\frac{\sigma_{dc}}{\omega \varepsilon_{0}}$ in Equation (Equation 5), it is necessary to remove the effect of the term from the $\varepsilon_{r}^{\prime \prime }\left(\omega \right)$ for the exposure of the actual β-dispersion characteristic, using Equation (Equation 6) as follows: (6)$$\varepsilon_{rd}^{\prime \prime }\left(\omega \right)=\varepsilon_{r}^{\prime \prime }\left(\omega \right)-\frac{\sigma_{dc}}{\omega \varepsilon_{0}}$$

Using Equation (Equation 6), the curve-fitted and measured $\varepsilon_{rd}^{\prime \prime }\left(\omega \right)$ values were calculated and compared to each other (Figure 6b). The $\sigma_{dc}$ values of the protein suspensions of hMSCs and Saos-2 cells, given in Table 3, were used in these calculations.

Although there was a slight deviation between the curve-fitted and measured $\varepsilon_{rd}^{\prime \prime }\left(\omega \right)$ below 100 MHz, the curve-fitted $\varepsilon_{rd}^{\prime \prime }\left(\omega \right)$ tends to follow the measured $\varepsilon_{rd}^{\prime \prime }\left(\omega \right)$. The $\varepsilon_{rd}^{\prime \prime }\left(\omega \right)$ spectra of the protein suspensions of hMSCs and Saos-2 cells (Figure 6b) reveal the difference in $\varepsilon_{rd}^{\prime \prime }\left(\omega \right)$ values at the relaxation frequency ($f_{c,\beta }$). At $f_{c,\beta }$ of the protein solutions of hMSCs and Saos-2 cells (Table 3), the difference in $\varepsilon_{rd}^{\prime \prime }\left(\omega \right)$ values between the protein suspensions of hMSCs and Saos-2 cells was 28.16. The difference in $f_{c,\beta }$ between the protein suspensions of hMSCs and Saos-2 cells is 723 kHz. However, in the γ-dispersion region, the $\varepsilon_{rd}^{\prime \prime }\left(\omega \right)$ spectra are barely affected by the term, $\frac{\sigma_{dc}}{\omega \varepsilon_{0}}$, as expected.

#### 3.2.3. Dielectrophoresis Study

DS can provide complex permittivity spectra over a broadband frequency range but is limited below the MHz frequency band. Another label-free technique called dielectrophoresis (DEP) is used to discriminate between cells at different stages of differentiation. Muratore et al. demonstrated that DEP can discriminate cells between the stages of differentiation in a C2C12 myoblast multipotent mouse model [46]. Velugotla et al. used DEP to discriminate between undifferentiated human embryonic stem cell lines and differentiated progenies [47]. Both studies used a low frequency (kHz range) to acquire the DEP response.

The DEP cross-over frequency, $f_{\mathrm{xo}}$, is the primary parameter in DEP. The $f_{\mathrm{xo}}$ for a spherical cell with radius *R* is given to good approximation by
(7)$$f_{\mathrm{xo}}=\frac{\sqrt{2}\;\sigma_{m}}{2\pi \;R\;C_{m}}$$
where $\sigma_{m}$ is the conductivity of the suspending medium, and $C_{m}$ is the specific capacitance (F/m) of the cell membrane. The $C_{m}$ is related to the effective permittivity, $\varepsilon_{\mathrm{eff}}$, of a spherical cell with radius *R* as follows: (8)$$\varepsilon_{\mathrm{eff}}=\frac{\varepsilon_{0}\;\varepsilon_{m}\;R}{\delta }\;\phi_{m}=R\;C_{m}$$
where δ is the membrane thickness, and $\varepsilon_{m}$ is the mean relative permittivity of the material forming the membrane structure. Factor $\phi_{m}$ in Equation (Equation 8) is termed the membrane-folding factor, and $\phi_{m}=1$ for a perfectly smooth spherical cell [46].

DEP is a technique that is applied to cell suspensions. If the proteins of hMSCs and Saos-2 cells can be modeled as spherical particles, it may be applied to the proteins of hMSCs and Saos-2 cells. The $\tau_{\beta }$ of a spherical protein of hydrodynamic radius *R* in a viscous medium η can be expressed as follows [44,48]: (9)$$\tau_{\beta }=\frac{4\;\pi \;R^{3}\;\eta }{k_{B}\;T}$$
where $k_{B}$ is the Boltzmann constant, and *T* is the absolute temperature. In Section 3.2.2, $\tau_{\beta }$ is determined by using the curve-fitting technique. Therefore, the hydrodynamic radius *R* of a spherical protein can be calculated by using Equation (Equation 9).

According to Schwan [45], the relative permittivity of “bound water”, $\varepsilon_{\mathrm{bw}}$, can be determined using the following Equation (Equation 10): (10)$$\frac{\varepsilon_{e}\;-\;\varepsilon_{\mathrm{bw}}}{\varepsilon_{e}\;+\;2\;\varepsilon_{\mathrm{bw}}}\;={\left(\frac{R}{R+d}\right)}^{3}\frac{\varepsilon_{p}\;-\;\varepsilon_{\mathrm{bw}}}{\varepsilon_{p}\;+\;2\;\varepsilon_{\mathrm{bw}}}$$
where $\varepsilon_{e}$ is the effective permittivity of a hydrated protein in a spherical shape, *d* is the thickness of the shell of “bound water”, and $\varepsilon_{p}$ is the assumed relative permittivity of the protein. The $\varepsilon_{e}$ in Equation (Equation 10) can be determined using the following Equation (Equation 11): (11)$$\frac{\varepsilon_{\mathrm{sus}}\;-\;\varepsilon_{\mathrm{med}}}{\varepsilon_{\mathrm{sus}}\;+\;2\;\varepsilon_{\mathrm{med}}}\;=\;p\frac{\varepsilon_{e}\;-\;\varepsilon_{\mathrm{med}}}{\varepsilon_{e}\;+\;2\;\varepsilon_{\mathrm{med}}}$$
where $\varepsilon_{\mathrm{sus}}$ is relative permittivity of the protein suspension, $\varepsilon_{\mathrm{med}}$ is relative permittivity of the suspending medium, and *p* is the volume fraction of hydrated protein.

The $\varepsilon_{m}$ and δ in Equation (Equation 8) are equivalent to $\varepsilon_{\mathrm{bw}}$ and *d* in Equation (Equation 10), respectively. The volume of the hydrated protein is approximately 50 % more than that of the unhydrated one [45]. The value for *d* in Equation (Equation 10) is thus calculated using $d=\sqrt[{3}]{1.5}\;R$. The $\varepsilon_{\mathrm{sus}}$ of the protein suspensions of hMSCs and Saos-2 cells was measured, which enabled the $\varepsilon_{e}$ of the hydrated proteins of the hMSCs and Saos-2 cells to be determined using Equation (Equation 11).

Equations (Equation 7)–(Equation 11) can be used to determine the $f_{\mathrm{xo}}$ of the proteins from the hMSCs and Saos-2 cells. In these calculations, $\sigma_{m}=0.0032\;S/m$, $\eta =0.9\times 10^{-3}\;\mathrm{Pa}\;\cdot \;s$ at T=298K are used, and the *p*-value in Equation (Equation 11) is 0.411 when assuming the hydration value of protein to be 0.3 g.$H_{2}O$/g.protein [45]. The $\varepsilon_{p}$ in Equation (Equation 10) is assumed to be $\left(\varepsilon_{e}-\varepsilon_{\mathrm{sus}}\right)$. The results are shown in Table 4.

## 4. Discussion

IHC is widely used to analyze proteins and molecules in cells and tissues. This reaction usually occurs between two protein macromolecules. Glycoprotein antibodies bind to glycoproteins, lipoproteins, or protein antigens. The chemical constitution of an antibody determines its specific binding to a single antigen [49]. However, IHC analysis has limitations. One major limitation of stem cell assay is the lack of all markers necessary to distinguish them from other cells. IHC requires a multi-step process and takes time to process multiple steps. IHC can be performed on fixed cells or tissues following treatment to permit the entry of antibodies. The secondary antibody was then bound to the primary antibody for detection. Secondary antibodies were labeled with fluorophores and observed under a fluorescence microscope. To prevent non-specific antibody binding in the sample, various parameters were considered, such as antibody design, concentration, and treatment time; buffer constituents and pH; and temperature [50,51]. In addition, the fixation method may mask epitopes, thereby preventing the primary antibody from binding to its target. An antigen retrieval step is often performed to overcome this problem; pretreatment of the tissue retrieves the antigens masked by fixation, making them more accessible to antibody binding. Efficient antibody design is imperative because nonspecific antibodies create more background noise through off-target binding, preventing accurate data from being obtained [52]. These IHC complexities must be considered for each antibody and sample type before testing can be performed.

The self-renewal capacity of stem cells and their ability to differentiate into tissue-specific cells render them useful for regenerating tissue lost through disease or injury. Cocktails of various growth factors in the media, mechanical stimulation, and a specific extracellular matrix are required for their differentiation [53,54]. Stem cells treated with these differentiation conditions have previously been shown to upregulate the expression of several early and mature markers of tissue-specific cells [55,56].

Undifferentiated stem cells could cause tumor formation or differentiate into other cells [57,58]. Therefore, it is important to monitor the stem cell differentiation process through various analytical methods.

In this study, we evaluated the viability of DS to distinguish between hMSCs and mature Saos-2 cells. The complex spectra $\varepsilon_{r}\left(\omega \right)$ of the protein suspensions of hMSCs and Saos-2 cells provided by DS show two distinctive dispersions, called β- and γ-dispersions. The β-dispersion occured because of interfacial polarization between the protein and DW. Interfacial polarization occurs between dissimilar materials; the larger the difference in the dielectric properties between the two materials, the larger the interfacial polarization [12].

The first factor for detecting stem cell differentiation is the different $\varepsilon_{r}^{\prime }\left(\omega \right)$ spectra in the β-dispersion. A β-dispersion was observed in both the protein suspensions of hMSCs and Saos-2 cells, but $\varepsilon_{r}^{\prime }\left(\omega \right)$ of the protein suspension of hMSCs was greater than that of the protein suspension of Saos-2 cells below 300 MHz. This implied that the interfacial polarization in the protein suspension of hMSCs was larger than that in the protein suspension of Saos-2 cells. Therefore, the $\varepsilon_{r}^{\prime }\left(\omega \right)$ value of hMSCs in the low-frequency band was greater than that of Saos-2 cells because both protein types were suspended in DW.

The second factor for detecting stem cell differentiation is the different $\varepsilon_{rd}^{\prime \prime }\left(\omega \right)$ spectra in the β-dispersion. The $\varepsilon_{r}^{\prime \prime }\left(\omega \right)$ of the protein suspension of hMSCs was greater than that of the protein suspension of Saos-2 cells below 1 GHz. When comparing the $\varepsilon_{rd}^{\prime \prime }\left(\omega \right)$ of the protein suspension of hMSCs to that of the protein suspension of Saos-2 cells at $f_{c,\beta }$, the $\varepsilon_{rd}^{\prime \prime }\left(\omega \right)$ of the protein suspension of hMSCs was 28.16 greater than that of the protein suspension of Saos-2 cells. From the curve-fitted parameters, it can be seen that $\sigma_{dc}$ of the protein suspension of hMSCs was 20 % greater than that of the protein suspension of Saos-2 cells.

The third factor for detecting stem cell differentiation is the different $f_{c,\beta }$ in the β-dispersion. Using the curve-fitting technique, the difference in $f_{c,\beta }$ between the protein suspensions of hMSCs and Saos-2 cells was found to be 723 kHz, which is a distinguishable value in the low frequency band.

In the “single-shell” modeling, there was a little discrepancy between the measured and curve-fitted model in $\varepsilon_{rd}^{\prime \prime }\left(\omega \right)$ spectra below 100 MHz. In the “single-shell” model, the suspended particles are assumed to be spherical shells. However, the proteins of hMSCs and Saos-2 cells cannot be spherically shaped particles, nor can they be the same size. Thus, the discrepancy between the measured and curve-fitted model in $\varepsilon_{rd}^{\prime \prime }\left(\omega \right)$ spectra below 100 MHz might be due to the difference in the shape of the particles in the mathematical model and the actual sample.

The DEP study for hydrated proteins was based on the model suggested by Schwan [45], who verified the model using a hemoglobin suspension. Although the proteins were different, the DEP cross-over frequencies, $f_{\mathrm{xo}}$, for the protein suspensions of hMSCs and Saos-2 cells were calculated, and the difference in $f_{\mathrm{xo}}$ was 4.26 kHz. This study demonstrates the relationship between DS and DEP, which must be verified using DEP measurements in future work.

We also evaluated the different cell types by immunohistochemistry using representative antibodies (a kind of protein), including CD 90, CD 73, and osteocalcin, which are present in hMSCs and Saos-2 cells. The changed dielectric profile using DS-sensing technology was due to the presence of other proteins in stem cells, including CD 90 and CD 73.

In this study, we evaluated and confirmed the viability of DS to distinguish hMSCs from mature Saos-2 cells using the protein suspensions of hMSCs and Saos-2 cells. In future work, hMSC and Saos-2 cell suspensions themselves and mixed cell suspensions should be investigated using DS. In addition to DS with cell suspensions, the relationship between DS and DEP should be studied to obtain a complete picture of the dielectric properties of cell suspensions down to the kilohertz frequency band. In addition to this, the concentration dependence of the complex permittivity spectra must be studied to investigate the sensitivity of DS for sensing stem cell differentiation. All experiments must be reproducible, and appropriate statistical analyses should be conducted.

## 5. Conclusions

In this study, we demonstrated that dielectric spectroscopy is a novel technique for sensing stem cell differentiation. The complex permittivity spectra of the protein suspensions of hMSCs and Saos-2 cells were compared in the frequency range of 10 MHz to 43.5 GHz. The real ($\varepsilon_{r}^{\prime }$) and imaginary ($\varepsilon_{r}^{\prime \prime }$) parts of the complex permittivity spectra of the protein suspension of hMSCs were distinguishable from those of the protein suspension of Saos-2 cells below 300 MHz and 1 GHz, respectively. The differences in $\varepsilon_{r}^{\prime }$ and $\varepsilon_{r}^{\prime \prime }$ are two distinctive features that indicate stem cell differentiation. In the complex permittivity spectra, two major dispersions, β and γ, were observed. The difference in relaxation frequency in β-dispersion is an additional feature to detect stem cell differentiation. Because dielectric spectroscopy is noninvasive, label-free, and capable of real-time sensing, it is a promising sensing technique to detect stem cell differentiation. In the dielectrophoresis study for the proteins of hMSCs and Saos-2 cells, we showed that protein dielectrophoresis response can be obtained by using the β-dispersion parameters in the dielectric spectroscopy measurements, which confirmed that protein dielectrophoresis can discriminate the proteins from hMSCs and Saos-2 cells.

## Figures and Tables

**Figure 1 sensors-23-02397-f001:**
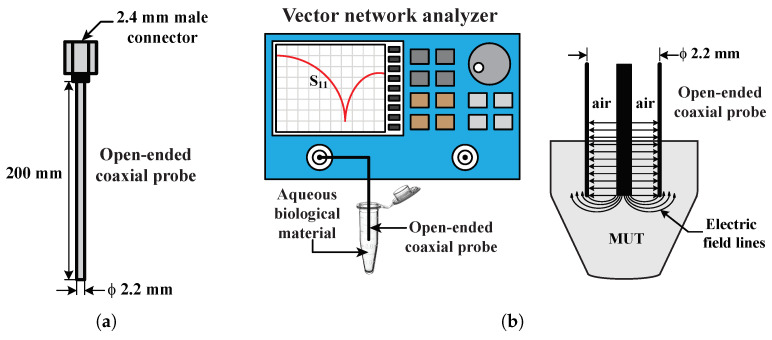
Schematics of (**a**) an open-ended coaxial probe and (**b**) RF/microwave band measurement configuration for aqueous biological material with an open-ended coaxial probe [19].

**Figure 2 sensors-23-02397-f002:**
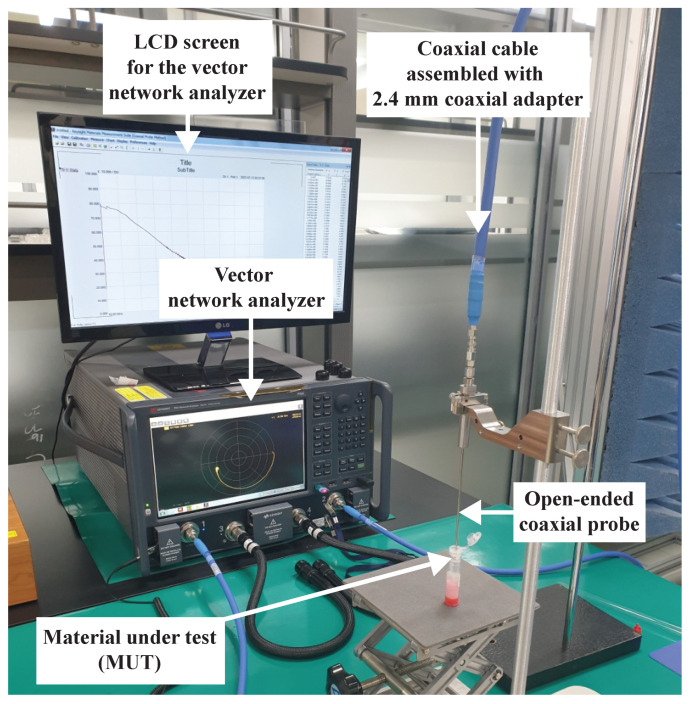
Measurement configuration for the material-under-test.

**Figure 3 sensors-23-02397-f003:**
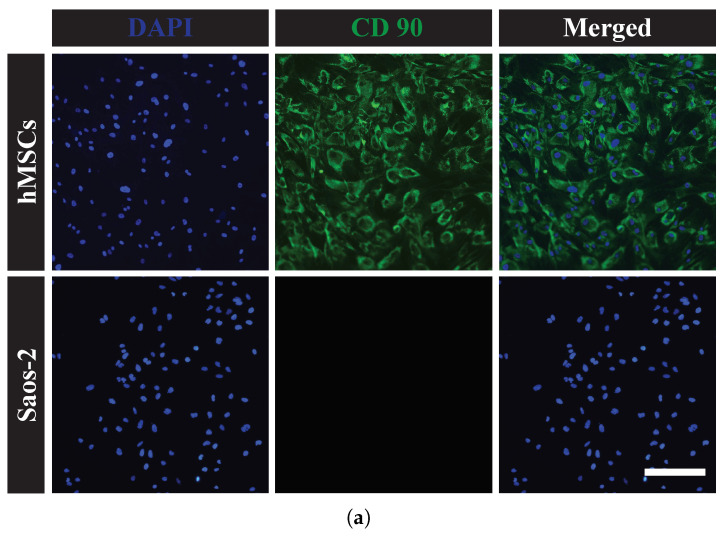
Immunofluorescence staining for: (**a**) CD 90; (**b**) CD 7; and (**c**) osteocalcin in hMSCs and Saos-2 cells. The scale bars indicate 200 μm. Green staining indicates CD 90, CD 73, or osteocalcin, and blue staining indicates cell nuclei by 4’,6-diamidino-2-phenylindole (DAPI) staining.

**Figure 4 sensors-23-02397-f004:**
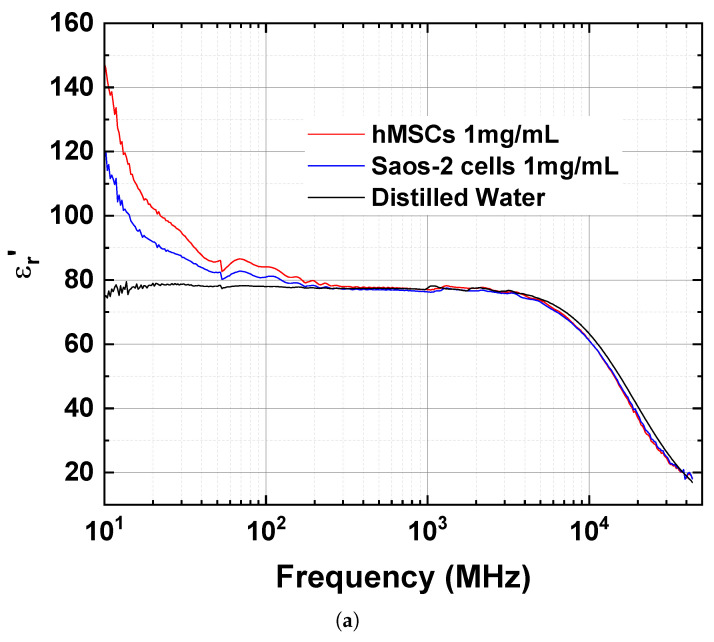
Comparison of (**a**) real and (**b**) imaginary part of the permittivity spectra for the protein suspensions of hMSCs and Saos-2 cells.

**Figure 5 sensors-23-02397-f005:**
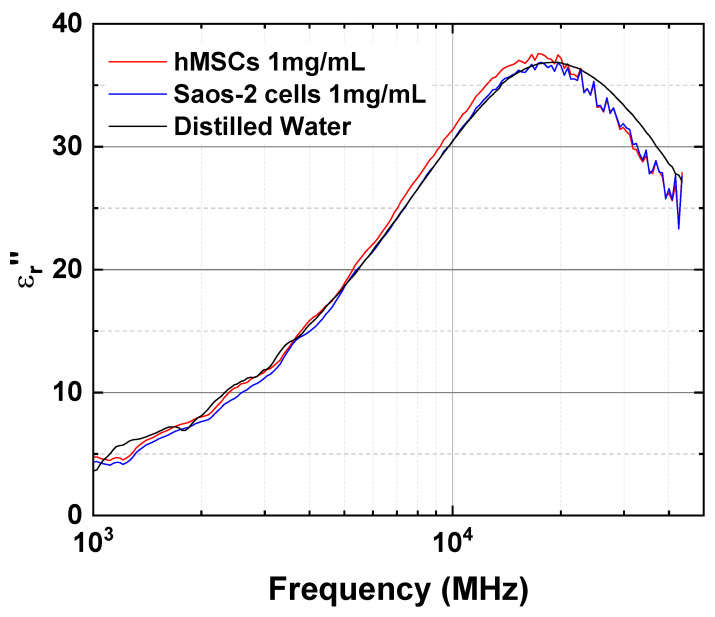
Imaginary part of the complex permittivity spectra focused on the microwave frequency range.

**Figure 6 sensors-23-02397-f006:**
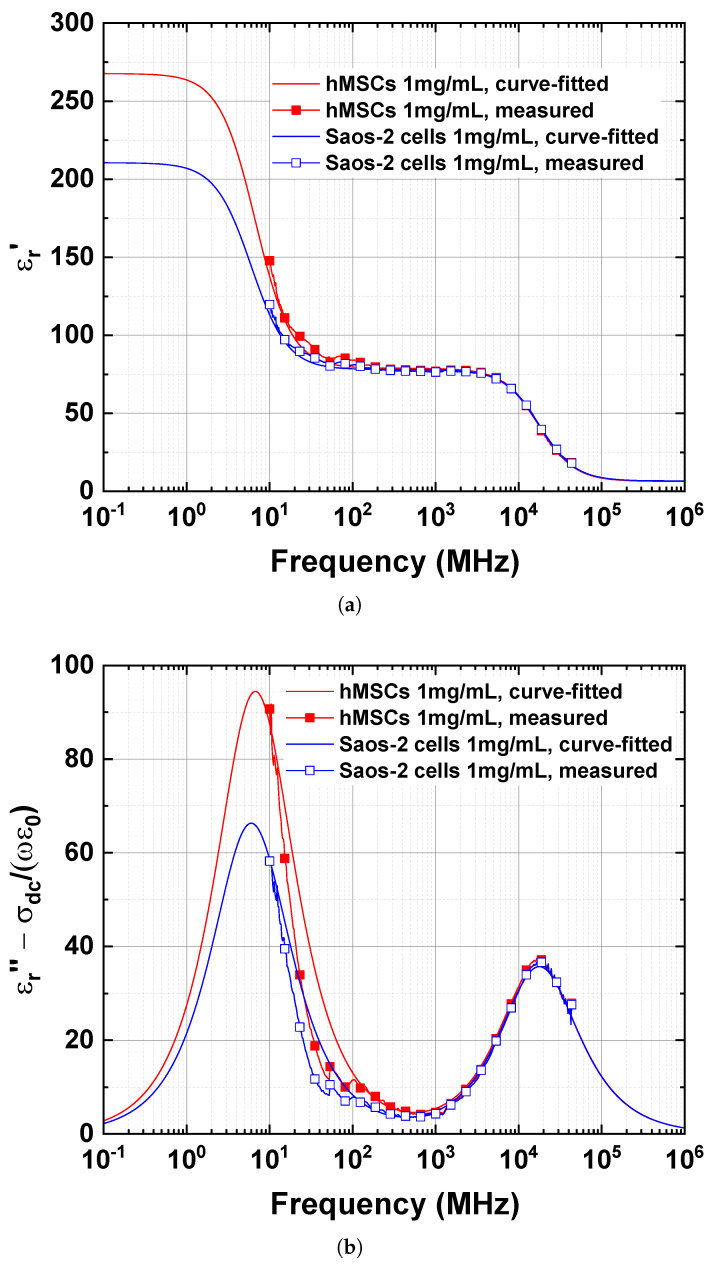
Comparison of (**a**) real and (**b**) imaginary part of the permittivity spectra between the curve-fitted and measured results for the protein solution of hMSCs and Saos-2 cells.

**Table 1 sensors-23-02397-t001:** Comparison of measured $\varepsilon_{r}^{\prime }$ and $\varepsilon_{r}^{\prime \prime }$ of the three materials at specific frequencies.

Frequency	$\varepsilon_{r}^{\prime }$			$\varepsilon_{r}^{\prime \prime }$		
(MHz)	Suspension of hMSCs	Suspension of Saos-2 Cells	Distilled Water	Suspension of hMSCs	Suspension of Saos-2 Cells	Distilled Water
10	147.7	119.6	74.3	108.6	68.1	14.5
30	94.4	87.0	78.7	28.7	17.9	4.3
300	77.9	77.1	77.4	6.4	4.6	2.0
1000	77.0	76.3	77.5	4.8	4.4	3.6
3000	76.2	75.8	76.5	11.9	11.4	11.9

**Table 2 sensors-23-02397-t002:** Comparison of relaxation frequencies and relaxation times in γ-dispersion among the three materials.

Material	$f_{c,\gamma }$ (GHz)	$\tau_{\gamma }$ (ps)
	(Relaxation Frequency)	(Relaxation Time)
Suspension of hMSCs	17.3	9.20
Suspension of Saos-2 cells	17.7	8.99
Distilled water	19.6	8.12

**Table 3 sensors-23-02397-t003:** Comparison of “single-shell” model parameters in Equation (Equation 4) between the protein suspensions of hMSCs and Saos-2 cells.

Material	$\varepsilon_{r,\infty }$	$\Delta \varepsilon_{\beta }$	$\tau_{\beta }$	$f_{c,\beta }$	$\Delta \varepsilon_{\gamma }$	$\sigma_{\mathrm{dc}}$
			(ns)	(MHz)		(S/m)
Suspension of hMSCs	6.364	188.865	23.680	6.721	72.435	0.01666
Suspension of Saos-2 cells	6.594	132.555	26.535	5.998	71.381	0.01333

**Table 4 sensors-23-02397-t004:** Comparison of DEP parameters between the proteins of hMSCs and Saos-2 cells.

Material	*R*	*d*	$\varepsilon_{\mathrm{bw}}$	$C_{m}$	$f_{\mathrm{xo}}$
	(Å)	(Å)		(F/m)	(kHz)
hMSCs protein	20.5	2.97	786.8	23.5	14.95
Saos-2 protein	21.3	3.08	613.9	17.6	19.21

## Data Availability

Data available on request from the authors.

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
