# Peer review of "Novel Sensing Technique for Stem Cells Differentiation Using Dielectric Spectroscopy of Their Proteins"

_sensors, 2023, doi:10.3390/s23052397_

Round 1
Reviewer 1 Report
This is a well written and presented paper that reports careful measurements of the complex permittivity of suspensions of human mesenchymal stem cells and osteogenic sarcoma cells. A good summary of published and relevant dielectric studies of bioparticles is also presented. Although the experimental technique used by the authors is a standard one, described often in the literature, their choice of cells to study is novel.
The main conclusion stated by the authors is that their kind of application of dielectric spectroscopy can be expanded to detect the differentiation of stem cells. However, although their experimental results support this conclusion, the manuscript has two major deficiencies which should be corrected.
1.
No efforts have been made by the authors to analyze their data by means of ‘multishell’ models of a cell that can provide details of what dielectric properties of a cell are changing during differentiation (e.g., cell membrane capacitanec, conductance or changes of cytoplasm dielectric properties).
Some key papers to cite, study and apply to their own work are:
Irimajiri, A., Hanai, T. and Inouye, A. (1979) Dielectric theory of multi-stratified shell-model with its application to a lymphoma cell, J. Theor. Biol. 78(2), 251-269.
Irimajiri, A., Asami, K., Ichinowatari, T., et al., (1987) Passive electrical properties of the membrane and cytoplasm of cultured rat basophil leukemia cells: I. Dielectric behaviour of cell suspensions in 0.01-500 MHz and its simulation with a single-shell model, Biochim. Biophys. Acta, 896(2), 203-213.
Irimajiri, A., Doida, Y., Hanai, T., et al., (1978) Passive electrical properties of cultured marine lymphoblasts (L5178Y) with reference to its cytoplasmic membrane, nuclear-envelope and intracellular phases, J. Membrane Biol., 38(3), 209-232.
2.
The authors also appear to be unaware of an extensive literature that has already demonstrated that the differentiation of various types of cells (including human mesenchymal stem cells) can be monitored through changes of their complex permittivity values, as observed in dielectrophoresis (DEP) experiments. Because of this, the authors have failed to identify a significant significance of their results in validifying some of this DEP work.
Two papers of relevance to cite, study and incorporate into their thinking are:
Muratore, M., Srsen, V., Waterfall, M., et al., (2012) Biomarker-free dielectrophoretic sorting of differentiating myoblast multipotent progenitor cells and their membrane analysis by Raman spectroscopy, Biomicrofluidics 6, 034113.
Velugotla, S., Pells, S., Mjoseng, H. K., et al (2012) Dielectrophoresis based discrimination of human embryonic stem cells from differentiating derivatives, Biomicrofluidics 6, 044113.
Because the authors are not aware of this literature, it can surmise that they are newcomers to the field of biodielectric spectroscopy and so not aware of the full impact that they are capable of achieving in this field. To workers in this field, a typical response to this paper might be: ’So what. Who cares?’ This would be unfortunate. I strongly believe that the authors can raise the impact of this paper by at least citing the multi-shell model of a cell, coupled with attempts to apply them. Also, their dielectric spectroscopy measurements act to confirm DEP date. Adding DEP measurements to their DS work would bring a new dimension to the field. They should at least make reference to the relevance of DEP to their (hopefully) long-term aims and objectives in their research. Better still, they should be able to use their own results on mesenchymal cells to explain if they support the DEP studies. This kind of analysis is missing in the DEP literature.
I hope they see this as encouragement to them – rather than a criticism. I think they are capable of great discoveries and important biomedical applications.
Author Response
Enclosed, please find the response letter for the reviewer's comments.

Reviewer 2 Report
In this manuscript (sensors-2194571), dielectric spectroscopy was reported as a novel technique for sensing stem cell differentiation sensitively in broadband frequency range compared with immunohistochemistry. Despite novelty of this work, it needs to be major revised before it can be published on Sensors.
Some detail information need to be clarified :
1. In the introduction section, it should be more summative and briefer about the description of DS development process rather than listing literature without analysis by synthesis. The author should rewrite it.
2. The amount of experiment works that examine whether hMSCs can be distinguished from Saos-2 cell suspensions by the complex permittivity spectra is insufficient. Firstly, the conclusion and equation model should be drew after reproducible experiments and statistical analysis. Secondly, the complex permittivity spectra detected under mixed cell suspensions of hMSCs and Saos-2 should be added into experiment which is more convincing.
3. In the comment text in Figure 4, the spelling of Saos-2 is incorrect. Please correct it.
4. The levels of CD90, CD73, and osteocalcin may change during the differentiation of stem cells, depending on the stage of differentiation. Does this change lead to a change in its dielectric profile? Does the ability of dielectric spectroscopy to detect such changes have an advantage over traditional immunohistochemistry? Please further discuss or look forward to it in the paper.
Author Response

(The authors gave the same response as above.)

Round 2
Reviewer 1 Report
Now that I understand that DS measurements were performed on suspensions of proteins and NOT cells, this work is now of much greater novelty and importance.
The title of the paper is still misleading - wording such as 'of their proteins' should be added. The fact that DS of proteins is studied should be EMPHASISED. The reason why this work is of much larger significance, in my view, is that for 30 years there has been a great mystery regarding the DEP of proteins. Labs capable of performing DS measurements of proteins now hold a major key for advancing protein DEP into novel biosensors and pharmaceutical applications, for example. This is summarized by Colborn & Matyushov in a very recent preprint
https://chemrxiv.org/engage/chemrxiv/article-details/63da7edd33b6976c48030cbc and also reviewed by Holzel et al (Electrophoresis 2021, 42, 513-538) and Pethig (Micromachines 2022, 13, 261).I do not think the authors should extend their current manuscript too much to include the important relevance of their work to protein DEP - just a few lines and references to add into the Introduction and Concluding comments to emphasise and answer the 'So what? Who cares?' questions. I know of two labs seeking large funding to develop DS of proteins coupled to the DEP of proteins. In my view the authors could also be major players in this activity. A few minor adjustments to this present paper would add an entry stamp for this :)
Author Response

(The authors gave the same response as above.)

Reviewer 2 Report
The revised manuscript has been improved. For the questions about this work, the authors have made careful revisions and supplements and added a relevant and reliable subsection, and the responses are reasonable. Thus, I would like to recommend this paper to be published in Sensors.
Author Response

(The authors gave the same response as above.)
